# Effect of Dietary Fatty Acids on MicroRNA Expression Related to Metabolic Disorders and Inflammation in Human and Animal Trials

**DOI:** 10.3390/nu13061830

**Published:** 2021-05-27

**Authors:** Karla MacDonald-Ramos, Alejandra Martínez-Ibarra, Adriana Monroy, Juan Miranda-Ríos, Marco Cerbón

**Affiliations:** 1Unidad de Investigación en Reproducción Humana, Instituto Nacional de Perinatología “Isidro Espinosa de los Reyes”-Facultad de Química, Universidad Nacional Autónoma de México, Ciudad de México 11000, Mexico; kmmacdonald@yahoo.com (K.M.-R.); alejandra_martinez@hotmail.es (A.M.-I.); 2Doctorado en Ciencias Biológicas y de la Salud, Universidad Autónoma Metropolitana, Ciudad de México 04960, Mexico; 3Servicio de Oncología, Hospital General de México Dr. Eduardo Liceaga, Ciudad de México 06720, Mexico; adriana_monroy_guzman@hotmail.com; 4Unidad de Genética de la Nutrición, Departamento de Biología Molecular y Biotecnología, Instituto de Investigaciones Biomédicas de la Universidad Nacional Autónoma de México, Ciudad de México 04530, Mexico; riosjuanm@iibiomedicas.unam.mx; 5Instituto Nacional de Pediatría, Ciudad de México 04530, Mexico

**Keywords:** dietary fatty acids, polyunsaturated fatty acids, monounsaturated fatty acids, miRNA expression, metabolic disorders, cardiovascular disease, inflammation

## Abstract

Dietary fatty acids (DFAs) play key roles in different metabolic processes in humans and other mammals. DFAs have been considered beneficial for health, particularly polyunsaturated (PUFAs) and monounsaturated fatty acids (MUFAs). Additionally, microRNAs (miRNAs) exert their function on DFA metabolism by modulating gene expression, and have drawn great attention for their potential as biomarkers and therapeutic targets. This review explicitly examined the effects of DFAs on miRNA expression associated with metabolic diseases, such as obesity, non-alcoholic fatty liver disease (NAFLD), and cardiovascular disease (CVD), as well as inflammation, published in the last ten years. DFAs have been shown to induce and repress miRNA expression associated with metabolic disease and inflammation in different cell types and organisms, both in vivo and in vitro, depending on varying combinations of DFAs, doses, and the duration of treatment. However, studies are limited and heterogeneous in methodology. Additionally, recent studies demonstrated that high fat ketogenic diets, many enriched with saturated fats, do not increase serum saturated fat content in humans, and are not associated with increased inflammation. Thus, these findings shed light on the complexity of novel treatment and DFA interventions for metabolic disease and to maintain health. Further studies are needed to advance molecular therapeutic approaches, including miRNA-based strategies in human health and disease.

## 1. Introduction

Dietary fatty acids (DFAs) play crucial roles in different metabolic processes in humans and other mammals. DFAs are major fuel substrates that usually comprise one-third of the total energy intake in humans. When dietary intake outweighs energy expenditure, it may contribute to the development of metabolic disorders, such as obesity, NAFLD [1], and cardiovascular disease (CVD) [2,3], the major cause of death worldwide [2]. Particularly, CVD is associated with the intake of simple sugars and refined carbohydrates [4,5,6,7]. Moreover, studies have shown a lack of risk for CVD associated with dietary saturated fatty acids (SFAs) [8,9]. However, controversy exists, as food-based guidelines continue to recommend limiting the consumption of animal fats high in SFAs [10].

The American Heart Association (AHA) and the National Cholesterol Education Program recommend 25% to 35% of daily calories from fat, of which SFAs should only comprise 7%–10% of total energy intake, and consumption of TFAs (trans fatty acids; industrially produced and described below) should be kept as low as possible [11]. It is known that the linear structure of SFAs can alter cell membranes and signal inflammatory molecules. SFAs have been found to increase low-density lipoprotein (LDL) cholesterol, a major risk factor for developing obesity, and CVD [12]. Furthermore, TFAs also increase LDL cholesterol, as well as triglyceride levels, and reduce high-density lipoprotein cholesterol levels [13,14]. Additionally, the consumption of commercial partially hydrogenated vegetable oils increases the risk of developing coronary heart disease, metabolic syndrome, and type 2 diabetes (T2D). However, the substitution of SFAs with polyunsaturated fatty acids (PUFAs) in the diet appears to have favorable effects on LDL cholesterol. Moreover, it is recommended to replace animal fats in the diet with PUFAs and MUFAs and to limit the consumption of food products that contain synthetic sources of TFAs [11].

DFA structure influences gene expression, and will therefore be described below in greater detail. DFA structure varies widely in hydrocarbon chain length, ranging from two to 36 carbon atoms with different levels of saturation. DFAs may be classified according to the presence or absence of double bonds in SFAs, the presence of one double bond in MUFAs, and the presence of more than one double bond in PUFAs [15]. They may also be classified according to the position of the first double bond with the fatty acid methyl terminus. For example, in ω-3 DFAs, the first double bond is found at the third carbon of the chain from the methyl terminus. In ω-6 DFAs, the first double bond is found at the sixth carbon of the chain from the methyl terminus, and so on [11]. On the other hand, TFAs are another type of fatty acid formed industrially that also contain double bonds. They are secondary products of cis-double bonds interacting with certain catalysts used to partially hydrogenate high PUFA vegetable oils. Ruminant TFAs are also found in ruminant meat and milk resulting from the biohydrogenation of unsaturated fatty acids by bacterial metabolism found in the rumen of animals like cattle and goats [11,14,16].

Differences in chain length and the level of saturation define DFA function in metabolism, impact human health and disease [11], and alter gene expression differently [17]. Moreover, DFAs have been found to modify gene expression across different cell types. Regulation of gene expression is conducted by a group of mechanisms that include activation of transcription factors, classical epigenetic processes [3,18,19], and other mechanisms, like those exerted by non-coding RNAs, such as microRNAs (miRNAs). In addition to gene expression modification from DFAs, lipoproteins have also been found to induce miRNA-dependent regulation in obesity and CVD, such as ox-LDL (oxidized low-density lipoprotein) [20,21].

miRNAs are the RNA products of the transcription of non-coding segments of the genome that regulate the post-transcriptional organization of protein-coding RNAs [22]. Animal canonical miRNAs are encoded from monocistronic, polycistronic, or intronic gene sequences, and transcribed by RNA polymerase II in the nucleus of the cell to form capped and polyadenylated primary miRNAs (pri-miRNAs) with hairpin structures and 5′ and 3′ ends. They are then cleaved into single hairpins by a microprocessor complex comprised of Drosha, DGCR8, the double-stranded RNA, and other auxiliary factors, including DEAD-box (DEAD box proteins are characterized by the presence of an Asp-Glu-Ala-Asp (DEAD) motif) RNA helicases p68 and p72 to form precursor miRNAs (pre-miRNAs). For efficient cleavage of the pri-miRNA, DGCR8 binds at the apical loop of the hairpin, and Drosha interacts with the basal stem [23]. The pre-miRNAs are then exported to the cytoplasm through a nuclear pore by export receptor exportin 5, where they are further processed by Dicer and TRBP in humans. Dicer cleaves the pre-miRNA close to the terminal loop to form a dsRNA 20–25 nucleotides long, and then gives the duplex to a member of the Argonaute (AGO) family of proteins (AGO 1–4 in humans) in an ATP-dependent manner [24] in a process termed RISC loading. Then, AGO, loaded with the miRNA duplex, dissociates from Dicer. TRBP, AGO, and the mature miRNA duplex form the RNA-induced silencing complex (RISC) [23].

Either one of the strands from the mature miRNA duplex may be loaded into AGO, however, the name of the mature miRNA will depend on the directionality of the miRNA strand that is loaded. If the 5′ end of the pre-miRNA hairpin is loaded, it will give rise to a 5p strand. The same is true for the 3p strand if the 3′ end of the miRNA duplex is loaded. The proportion of 5p or 3p strands loaded for any miRNA will hinge on the cell type and cellular environment [24]. Once the mature miRNA is loaded into RISC, the miRNA guides the complex to the complementary sequence in the 3′ untranslated region of its target mRNA, through the “seed sequence” found at the 5′ end of the miRNA [22] between nucleotides 2 and 7. Gene silencing occurs through the inhibition of translation after the formation of the miRNA–mRNA interaction, yet the mRNA is stable during the early stages of gene silencing. Later, the short poly (A) tails of mRNAs are degraded. Once miRNA interacts with its target sequence, AGO recruits a member of the GW protein family that contains a silencing domain, which also interacts with the poly (A) binding proteins that stimulate deadenylation, decapping, and, finally, mRNA decay [23].

Alternative non-canonical pathways for miRNA biogenesis that do not require either Drosha or Dicer complexes have been established [25]. Further details of non-canonical pathways and miRNA biogenesis regulation can be found in a review by Treiber et al. [23]. There are also multiple isoforms of miRNAs, and they can vary in length or contain untemplated nucleotide additions or post-transcriptional base edits and termed isomiRs. They are generated by either trimming the 5′ end or 3′ end of miRNAs [26]. Details of sequence variants and nomenclature can be found in a review by Desvignes et al. [27]; miRNAs are also regulated by various mechanisms, and miRNA genes can change together with, or independently of, their host genes due to mutations or by methylation. Alternatively, they can be downregulated by Dicer and/or Drosha activity, which can be affected by endogenous compounds, such as hormones and cytokines, and exogenous chemicals, such as xenobiotics [28]. Finally, miRNAs can be found in extracellular fluids, acting as autocrine, paracrine, and/or endocrine regulators that modulate activities in the cell. Circulating miRNAs have been reported in plasma and serum, cerebrospinal fluid, saliva, breast milk, urine, tears, colostrum, peritoneal fluid, bronchial lavage, and seminal and ovarian follicular fluids. They are found in vesicles, such as exosomes, microvesicles, and apoptotic bodies associated with proteins, such as AGO2 [24].

Additionally, several dietary components, such as DFAs, exert an effect on miRNA expression profiles or their function [3]. To understand gene regulation, it is of particular importance to consider the role diet has in modulating these noncoding regulatory RNAs [29]. It has become clear that DFAs as bioactive compounds play a key role in modulating miRNA expression, which in turn participate in diverse physiological and pathological processes in human health and disease [30].

## 2. Materials and Methods

This is a retrospective study that explicitly examined the effects of DFAs on miRNA expression, published in the last ten years. The results of this review showed that the studies are diverse, and employed a wide array of methodologies. While some of the studies included interventions with only one type of fatty acid, others included more than one or many, combining either all types of DFAs, except for TFAs, or a single type of MUFA or PUFA. Moreover, some studies performed interventions in vivo and validated the results in ex vivo and in vitro models, while others only validated results in vivo. Despite the wide range of studies henceforth reviewed, the results were examined according to miRNAs that were particularly associated with metabolic disorders and inflammation. The data collected from the studies are summarized in Table 1 and discussed in detail below.

## 3. Results

### 3.1. DFAs Alter miRNA Expression Associated with Metabolic Disorders

It has been reported that immortalized human hepatocytes treated with oleic acid-induced steatosis represent a valuable model for the study of genetic and functional factors involved in the process of lipid accumulation and liver injury [47]. Interestingly, phosphatase and tensin homolog (PTEN) is a regulator of phosphoinositide 3-kinase signaling and an important tumor suppressor mutated/deleted in human cancers. PTEN deletion in the liver leads to insulin resistance, steatosis, inflammation, and cancer. It has been recently demonstrated that unsaturated fatty acids trigger steatosis by down-regulating PTEN expression in this model of hepatocytes via activation of a mammalian target of rapamycin (mTOR)/nuclear factor kappa B (NF-κB) complex. Additionally, unsaturated fatty acids upregulate the expression of miRNA-21, which binds to PTEN messenger RNA 3′-untranslated region and induces its degradation, involved in the physiopathology of steatosis [36] (Figure 1A, Hepatocyte).

In another high-fat diet (HFD) murine model, the expression in retroperitoneal adipose tissue of five miRNAs related to adipocyte differentiation (miR-143), lipid metabolism (miR-103 and 107), and obesity (miR-221 and 222), as well as the metabolic state of the animals, were modified in response to treatment with CLA (conjugated linoleic acids) at different doses. Furthermore, the same study showed that CLA treatments altered the expression levels of some adipogenic marker genes, such as glucose transporter type 4 (Glut4), lipoprotein lipase (LPL), peroxisome proliferator activator receptor gamma 2 (PPARc2), and CCAAT/enhancer-binding protein alpha (C/EBPa). Additionally, correlations between miRNAs and their genetic targets were found in adipocytes; at low dosage (3.0 or 10 mg CLA/animal for 37 days), miR-143 correlated positively with adiponectin and leptin expression; miR-103 correlated with two key markers of lipid metabolism, fatty acid synthase (FASN), and muscle carnitine palmitoyltransferase 1b (Cpt1b); miR-107 correlated with genes involved in fatty acid oxidation, such as uncoupling protein 2 (Ucp2) and Cpt1b, as well as with C/EBPa; miR-222 showed a significant correlation with genes related to adipocyte metabolism (Glut4), lipolysis (hormone-sensitive lipase (HSL), and patatin-like phospholipase domain containing 2 (Pnpla2)), lipogenesis (PPARc2, Fas, stearoyl-Coen-zyme A desaturase 1 (Scd1)), fatty acid oxidation (Ucp2), and adipocytokines, adiponectin, and tumor necrosis factor-alpha (TNF-α). miR-221 expression also correlated with TNF-α levels. Meanwhile, at high doses of CLA (3 or 10 mg CLA/animal for the first 30 days and 6 or 20 mg CLA/animal for the next 35 days), miR-107 expression in adipocytes was significantly correlated with LPL, PPARα, and TNF-α expression levels. Additionally, in animals fed both an HFD and a normal fat diet, a significant correlation was observed between miR-107 and Cpt1b expression levels. However, in both high-fat feeding and normal-fat feeding, CLA treatment maintained the correlation between miR-107 and Cpt1b expression. miR-221 also correlated positively with TNF-α and negatively with adiponectin in response to CLA treatment [41] (Figure 1A, Adipocyte).

Another animal model to evaluate the effects of fat intake on obesity conducted on Göttingen minipigs (Sus scrofa) demonstrated that the addition of 0.55% MUFAs and 6.86% PUFAs in a diet with lard for 11 weeks led to the downregulation of miR-122, but did not change the expression of cationic amino transport 1 (CAT1) in hepatocytes, which suggested that miR-122 was involved in obesity [38]. In another study, the expression of miR-467b and its direct target LPL were evaluated in a mouse NAFLD model and in vitro using the Hepa hepatocyte 1–6 cell line. The results showed that miR-467b expression was significantly downregulated in liver tissues of HFD-fed mice and steatosis-induced hepatocytes. This finding suggested that the downregulation of miR-467b expression was involved in the development of hepatic steatosis through the modulated expression of its target LPL. Moreover, the interaction between miR-467b and LPL gene expression was associated with insulin resistance, a major cause of NAFLD [37] (Figure 1A, Hepatocyte).

In a long-term model of obesity, after animals were fed an HFD for five months, the expression levels of miR-142-3p, miR-142-5p, miR-21, miR-146a, and miR-146b were induced, and miR-200b, miR-200c, miR-122, miR-133b, miR-1, miR-30a, miR-192, miR-193a-3p, and miR-30e expression levels were repressed in white adipose tissue. The validated miRNAs were associated with cell differentiation and metabolic processes related to the development of obesity [34]. In another rat model, DHA was administered to a group of dyslipidemic cafeteria diet-fed male Wistar rats, and both liver tissues and PBMCs were examined for miR-122 and miR-33a expression, both regulators of lipid metabolism in the liver. After being fed an HFD for ten weeks, both miR-122 and miR-33a expression levels were induced. PUFA treatment for the next three weeks repressed both miR-122 and miR-33a expression in the liver and even more so in PBMCs (Figure 1A, Hepatocyte and 1 B, Monocyte). Expression levels of FASN and Peroxisome proliferator-activated receptor beta/delta (Pparβ/*ƍ*), target genes for miR-122, and of carnitine palmitoyltransferase 1a (cpt1a) and ATP Binding Cassette Subfamily A Member 1 (abca1), target genes for miR-33a, were consistent with these findings, and correlated with a state of lipemia [31].

Another study assessed global changes in miRNA expression in beta cells in an obesity-associated T2D model. Animals were fed an HFD for eight weeks with cholesterol and lard. First, miRNA profiling was performed in isolated pancreatic islets and then validated in vitro in primary rat and human beta cells. Interestingly, changes in miR-132, miR-184, and miR-338-3p expression levels were observed in pre-diabetic mice, which suggested that changes in miRNA expression occur before the development of the disease. Afterward, upregulation of miR-132 was associated with increased expression of MAF BZIP Transcription Factor A (Mafa), a gene that plays an important role in the control of beta cell function and survival (Figure 1B, Pancreatic B cell). The results indicated that insulin resistance and obesity triggered dysregulation in miRNA expression, which allowed beta cells to first adapt to disease and sustain cell function. Later, gene expression was associated with the onset and manifestation of T2D, and finally with the demise of cells [35].

In a later study in cell lines, the authors assessed the beneficial effects of ω-3 long-chain fatty acids (FAs) in human hepatoma G2 (HepG2) and epithelial colorectal adenocarcinoma (Caco-2) cells. The results of the study indicated that the effects were exerted in part through the regulation of miRNA expression. The study demonstrated that different FA treatments result in different patterns of miRNA expression, involved both in lipid metabolism and cancer functions. Importantly, the differential miRNA expression depended on the type of FAs, but also on the pharmacological preparation of the treatment, such as micelles produced with oleic and palmitic acids, as well as DHA. In particular, miR-30c and miR-192 target genes related to lipid metabolism were characterized in transfection studies, and their function in gene expression was assessed. This study was one of the first to demonstrate that DFAs fine-tune gene expression regulation in a restrained and tissue-specific manner [39]. In a further study from the same research group, Caco-2 cells were treated with docosahexaenoic acid (DHA), conjugated linoleic acid (CLA), and cholesterol; the authors showed that miR-107 was differentially expressed by all treatments. The inhibition of miRNA-107 expression deregulates circadian rhythm and contributes to the pathophysiology of metabolic diseases, such as T2D and ASCVD (atherosclerotic cardiovascular disease), thus representing a possible therapeutic target for these diseases [40].

The effects of PUFAs on miRNA expression were assessed in a healthy population. The effect of DFAs in a PUFA-enriched diet, inferred from plasma fatty acid concentration, was linked to changes in circulating miRNAs. In the first experiment, 20 miRNAs were identified and differentially expressed in healthy women after consuming PUFAs in their diet. In a subsequent study, ten miRNAs were validated in both men and women in a larger group. Changes were seen in miRNA expression after eight weeks with daily walnut and almond intake [42]. In particular, miR-221 and CRP (high-sensitivity C-reactive protein) expressions were repressed after treatment with PUFAs.

In another study performed in a healthy population, the effects of a single dose of extra virgin olive oil (EVOO) were detected after four hours of intake. EVOO is a complex nutrient mix composed primarily of MUFAs (65.2%–80.8%). Other components include PUFAs (7.0–15.5%), tocopherols, squalene, pigments, and other compounds, such as phenolic compounds, triterpene dialcohols, and B-sitosterol [49]. Fourteen miRNAs were differentially modified, and seven were validated in peripheral blood mononuclear cells (PBMCs). The validated miRNAs have potentially beneficial effects on insulin resistance (i.e., miR-107), inflammation (i.e., miR-181b-5p, miR-23b-3p), and cancer (miR-19a-3p, miR-519b-3p), and may play an important role in preventing the onset of CVD and cancer [44].

A recent study aimed to analyze the effect of eicosapentaenoic acid (EPA) in brown adipose tissue (BAT) thermogenesis, which represents a target to promote weight loss in obesity. The effect of EPA treatment in mice after twelve weeks was analyzed in murine inguinal white and brown adipose tissue to demonstrate how brown fat and muscle cooperate to maintain normal body temperature. It has been shown that BAT can be stimulated to increase energy expenditure, and hence offers new therapeutic opportunities in obesity [50]. The study reported that treatment with EPA increased oxygen consumption, but, importantly, modulated free fatty acid receptor 4 (Ffar 4), a functional receptor for ω-3 PUFAs, which enhanced brown adipogenesis through the upregulation of miR-30b and miR-378, and activation of cyclic adenosine monophosphate (cAMP) [32] (Figure 1A, Adipocyte). In another study, both an HFD administered to mice for 14 weeks and palmitate treatment in HepG2 cells induced miR-96 expression in the liver of the mice and the cells; miR-96 directly targeted insulin receptor (INSR) and insulin receptor substrate 1 (IRS-1) and repressed gene expression post-transcriptionally. Additionally, the liver of HFD fed mice and the treated hepatocytes exhibited impairment of insulin-signaling and glycogen synthesis, revealing miR-96 as a promoter of hepatic insulin resistance pathogenesis in obese states [45] (Figure 1A, Hepatocyte).

Although the consumption of high quantities of TFAs associated with CVD risk and a decrease in HDL has been well-established, the effect of ruminant TFAs (rTFA, naturally occurring in ruminant meat and milk fat) on metabolic markers was reported until only recently. The next study analyzed the HDL-carried miRNA expression of two of the most abundant HDL-carried miRNAs found in plasma. After nine men were given a diet high in either industrial TFAs (iTFAs, produced industrially by partial hydrogenation of vegetable oils), or rTFAs, designed in a previous study that measured the effect of rTFAs on metabolic markers [51], miRNA expression was measured in blood plasma: miR-223-3p concentrations correlated negatively with HDL levels after a diet high in iTFAs, and positively with C-reactive protein levels after a diet high in rTFAs. Conversely, miR-135-3p concentrations correlated positively with both total triglyceride levels after a diet high in iTFAs, and with LDL levels after a diet high in rTFAs. The results suggest that diets high in TFAs, whether of industrial or ruminant origin, alter HDL-carried miRNA concentrations, and are associated with changes in CVD risk factors [13]. One year later, in a similar study model, the same group analyzed whether diets with different concentrations of iTFAs or rTFAs modify contributions of HDL-carried miRNAs associated with CVD to the plasmatic pool. HDL-carried miR-103a-3p, miR-221-3p, miR-222-3p, miR-376c-3p, miR-199a-5p, miR-30a-5p, miR-328-3p, miR-423-3p, miR-124-3p, miR-150-5p, miR-31-5p, miR-375, and miR-133a-3p varied in contribution to the plasmatic pool after diets high in iTFAS and rTFAs were given. The results showed that diets high in TFAs modify the HDL-carried miRNA profile, contribute differently to the plasmatic pool, and suggest that these miRNAs may be involved in the regulation of cardioprotective HDL functions [16].

In another NAFLD model, the effect of fish oil on miRNA expression in rat liver tissue was analyzed. Animals were either fed a diet rich in lard alone or supplemented with fish oil. The results from sequencing identified 79 miRNAs as differentially expressed in the group supplemented with fish oil compared to the control group. Importantly, the repressed expression of rno-miR-29c target-regulated the expression of Period Circadian Regulator 3 (Per3), a critical circadian rhythm gene associated with obesity and diabetes; rno-miR-328 expression was induced, as well as target-regulated Proprotein Convertase Subtilisin/Kexin Type 9 (Pcsk9), a gene that plays a crucial role in LDL receptor degradation and cholesterol metabolism. Finally, miR-30d expression was induced, and in turn regulated the expression of its target Suppressor of Cytokine Signaling 1 (Socs1), a gene involved in immunity and insulin and leptin signaling, respectively [33].

Finally, in an in vitro NAFLD model, HepG2 cells treated with free fatty acids (palmitic and oleic acids, 1:2) resulted in miR-26a repressed endogenous expression. Additionally, miR-26a overexpressed in cells induced a protective role on lipid metabolism and the progression of NAFLD [46] (Figure 1A, Hepatocyte).

The results of these studies also demonstrated that miRNA expression is modified by DFA intervention, which in turn regulates genes involved with obesity, NAFLD, CVD, and other metabolic diseases. Interventions with unsaturated FAs seem to have a beneficial effect on metabolism, while the opposite seems to occur with SFAs and TFAs.

### 3.2. DFAs Alter miRNA Expression Associated with Inflammation

The consumption of DFAs has traditionally been associated with inflammation, with the inflammatory response depending on the amount and type of DFAs consumed. However, controversy exists, as recent studies have also demonstrated that high fat and ketogenic diets with SFAs do not increase serum saturated fat content in humans and are not associated with increased inflammation [52]. In previous studies, it was reported that meals high in fat content promote translocation of endotoxins, mainly lipopolysaccharide (LPS) produced by the gut microbiome, into the bloodstream. LPS stimulates the inflammatory response, which may be both acute and chronic. SFAs may also induce inflammation by mimicking the actions of LPS [53]. Firstly, SFAs are an essential structural component of bacterial endotoxins [54]. Additionally, DFAs can promote endotoxin absorption [55]. Lastly, pro-inflammatory compounds in food may be absorbed, and prompt the inflammatory response directly by stimulating Toll-like receptors [56] found on cells of the innate immune system that lead to NF-kB activation and proinflammatory cytokine expression [57,58]. It has also been reported that, as postprandial lipemia is strongly correlated to endotoxin absorption, the addition of PUFAs and MFAs in the diet reduces LPS pro-inflammatory activity [53,59]. Diets high in SFAs have been shown to promote metabolic endotoxemia not only through changes in the microbiome and bacterial end products, but also in altering intestinal physiology and barrier function and enterohepatic circulation of bile [60].

In an inflammation rat model, animals were fed rat chow with added lard and corn oil, low in ω-3 PUFAs, to induce inflammation and mimic the common human diet. After the animals were divided into three groups, each group was either treated with ω-3 PUFAs, ω-6 PUFAs, or saline solution for 16 weeks. In the ω-3 treated rats, cluster of differentiation 8 T cells (CD8+ T) often and Treg populations significantly increased, and interleukin 6 (IL-6), C-reactive protein (CRP), and TNF-α expression levels decreased. This suggests that the ω-3 treated group was better able to reduce inflammation related to a diet low in ω-3. The opposite resulted in the ω-6 treated group. Moreover, obesity and a reduction in omental fat in the ω-3 treated group were significantly lower compared to controls. Later, miRNA expression was analyzed in serum, peripheral blood mononuclear cells (PBMCs), white adipose tissue, and hepatocytes. Importantly, in the ω-3 treated group, miR-146b-5p (directly targets TNF and is involved in TLR (Toll-like receptor) and TCR (T cell receptor) pathways) expression was significantly upregulated in serum compared to controls. While miR-19b-3p (directly targets PTEN and is involved in cell cycle and MAPK pathways) and miR-183-5p (directly targets PTEN and TNF and involved in MAPK pathways) expression levels in the ω-3 treated group were significantly upregulated in serum and PBMCs compared to controls. In the ω-3 treated group, miR-29c and miR-292-3p expression levels were upregulated in serum compared to controls, although not significantly. The results demonstrated that the expression levels for these miRNAs were regulated by PUFAs in the diet [43]. Altogether, these findings suggest that certain DFAs in the diet are capable of modulating miRNA expression related to increased inflammation and obesity, whereas PUFAs in the diet, particularly EPA and DHA, seem to do the opposite.

## 4. Conclusions

The overall data reported in this review indicate that DFAs play key roles in lipid metabolism by modulating miRNA expression. Importantly, both unsaturated fatty acids and SFAs may exert positive or negative effects in different cell types and organisms, both in vivo and in vitro, depending on varying combinations of DFAs, doses, and the duration of treatment. Notably, DFAs modulated miRNA expression related to metabolic parameters in human serum in both healthy individuals and patients with metabolic syndrome in as little as four hours [44]. Other data showed that DFAs modulated miRNA expression prior to the onset of disease [35], suggesting that miRNAs prime cells long before metabolic dysfunction, thus emphasizing the importance of miRNAs both as regulators of metabolism and possible therapeutic targets. However, studies are heterogeneous and limited concerning methodologies and models. Nevertheless, the published novel findings hold promise for the treatment of human metabolic disease and to maintain health by DFA interventions. Notwithstanding, further studies are needed that incorporate molecular approaches and biological systems together with omics technologies to add evidence to the reported data and advance therapeutic strategies in human health and disease.

## Figures and Tables

**Figure 1 nutrients-13-01830-f001:**
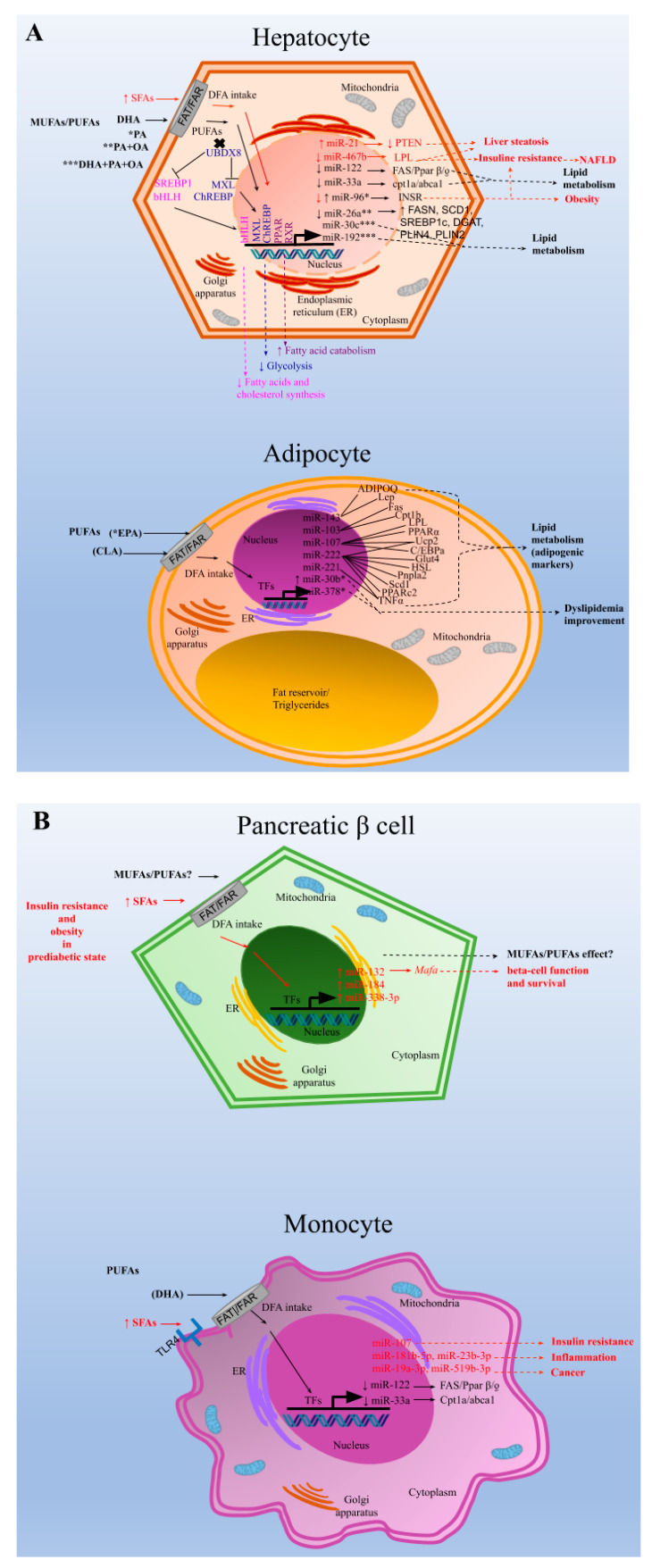
Dietary fatty acids (DFAs) regulate miRNA expression across different cell types. Regulatory mechanisms of miRNAs by (DFAs). Different treatments with monounsaturated fatty acids (MUFAs), as oleic or palmitic acid (PA), or polyunsaturated fatty acids (PUFAs), as docosahexaenoic ac (DHA) or eicosapentaenoic acid (EPA), or conjugated acid linoleic (CLA), used individually or in combination, leads to changes in specific microRNAs (miRNAs), either by inducing (↑) or repressing (↓) their expression, targeting genes and modulate them. DFAs get into cells through fatty acid transporters (FAT) or fatty acid receptors (FAR) (**A**) In hepatocytes, PUFAs reduce the expression of genes involved in fatty acid and cholesterol synthesis by binding and inactivating Ubiquitin regulatory X domain-containing protein 8 (UBXD8), thereby inhibiting the proteolytic processing of sterol binding protein (SREBP1). PUFAs also reduce the expression of L-type pyruvate kinase (glycolysis) in the liver, probably by inhibiting the nuclear translocation of MAX-like protein X (MLX)-carbohydrate responsive element binding protein (ChREBP). The activation of peroxisome proliferator activated receptor alpha (PPARα) by different PUFAs, or retinoid X receptor (RXR) specifically by DHA, leads to the stimulation of FAs catabolism [46]. Conversely, the SFAs induce miRNA changes which in turn, modulate the expression of target genes involved in the development of liver steatosis, insulin resistance, non-alcoholic fatty liver disease (NAFLD) and obesity. In adipocytes, EPA treatment it has shown modulated microRNAs associated with the improvement of dyslipidemia, whereas CLA treatment has shown to regulate the miRNAs expression, whose target genes are involved in lipid metabolism. (**B**) In pancreatic β cells, during a pre-diabetic state, high fat diet intake results in the deregulation of miRNAs related with function and survival of these cells. The effect of PUFAS/MUFAS has not been described in pancreatic cells. In Macrophages, Toll-like receptor 4 (TLR4) is present, and it has been proposed to be activated by saturated fatty acids (SFAs) [48]. In this cells, high fat diet intake results in changes of miRNAs expression related with insulin resistance, inflammation and cancer. DHA treatment have shown to modulate the expression of specific miRNAs and their target genes related with lipid metabolism. (**C**) In enterocytes, treatment with DHA or DHA+PA+oleic modulate the expression of miRNAs involved in lipid metabolism, while cellular intake of SFAs induces deregulation of miRNAs and their genes involved in the development of diseases such as Type 2 Diabetes (T2D), cardiovascular diseases (CVD), obesity and atherosclerosis. TFs was used to designate possible transcription factors not described or specified. Asterisks (*, **, or ***) were used to indicate the different treatments and their corresponding regulated miRNAs.

**Table 1 nutrients-13-01830-t001:** miRNAs modulated by DFAs and their effect on metabolic markers.

Type of DFAs	miRNAs	Type of Regulation	Organism/Cell/Tissue Type, Sample Size	Dose/Duration	Gene Expression	Ref
DHA	miR-33a, miR-122	repressed	Liver and PBMCs. Dyslipidemic cafeteria diet-fed male Wistar 150 g rats supplemented with DHA and/or proanthocyanidins. Five groups (*n* = 7)	Standard chow diet and cafeteria diet as a high-fat model. DHA: 515 mg PUFAs/kg (of body weight) dissolved in arabic gum, with sunflower lecithin and rosemary extract (flavoring) × 3 wks	qRT-PCR	[31]
EPA	miR-30b, miR193b, miR-365, miR-196a, miR-378	induced	Murine brown preadipocytes. iBAT from mice. C57BL/six male mice, (iBAT) adipocyte precursor cells, four diet groups (*n* = 8)	Murine primary brown adipogenic precursor cells from iBAT treated with 100 μM PA, OO, or EPA. Mice fed iso-caloric HF diet (50% cal from fat) with palm oil (PO), fish oil (FO), or olive oil (OO) × 12 wks.	qPCR	[32]
Fish oil, DHA, EPA	miR-345-5p, miR-34a-5p, miR-3556b, miR-3558-3p, miR-3590-5p, miR-362-3p, miR-374-3p, miR-374-5p, miR-455-5p, miR-466c-5p, miR-490-3p, miR-497-5p, miR-499-5p, miR-503-5p, miR-505-5p, miR-511-3p, miR-511-5p, miR-547-3p, miR-664-3p, miR-871-3p, miR-872-5p, miR-99a-3p, miR-9a-5p, let-7f-1-3p, miR-101a-5p, miR-106b-5p, miR-126b, miR-130a-3p, miR-142-3p, miR-142-5p, miR-144-3p, miR-144-5p, miR-146a-3p, miR-15b-3p, miR-17-5p, miR-18a-5p, miR-190a-5p, miR-193-3p, miR-19a-3p, miR-19b-3p, miR-22-5p, miR-223-3p, miR-23a-5p, miR-29b-3p, miR-29c-3p, miR-301a-3p, miR-32-5p, miR-33-5p, miR-330-5p, miR-331-3p, miR-339-5p	repressed	Nine-week-old male Sprague–Dawley rats, liver tissue	Lard-rich western diet (45 kcal% fat, 2% cholesterol) or Fish oil rich diet (45 kcal% fat and 2% cholesterol, 10% fish oil)	Illumina sequencing, RT-qPCR	[33]
	miR-100-5p, miR-10a-5p, miR-1249, miR-139-3p, miR-140-3p, miR-143-3p, miR-146b-5p, miR-148b-3p, miR-151-3p, miR-151-5p, miR-152-5p, miR-182, miR-203b-3p, miR-219-1-3p, miR-27b-5p, miR-28-5p, miR-293-5p, miR-30d-5p, miR-3102, miR-328a-3p, miR-3586-3p, miR-370-3p, miR-375-3p, miR-425-5p, miR-598-3p, miR-92a-3p, miR-99a-5p, miR-99b-5p	induced				
SFA (lard, cholesterol)	miR-342-3p, miR-146a miR-146b miR-222 miR-221 miR-142-3p miR-142-5p miR-21 miR-335-5p miR-146a miR-146b miR-647 miR-379	induced	White adipose tissue excised from the epididymal fat pad, C57BL6J wild type male mice eight weeks old fed HFD	Mice were fed HDF (60% fat, lard, and cholesterol) × five months.	microarray, qPCR	[34]
	MiR-141, miR-200a, miR-200a miR-200b miR-200c miR-122 miR-204 miR-133b miR-1 miR-30a miR-192 miR-193a-3p miR-203 miR-378 miR-30e	repressed				
HFD (cholesterol and lard)	miR-132, miR-199a-5p	induced	Dissociated islet cells from mice. Five wk old male C57BL/six mice and cell lines B-cells	Mice were fed an HFD (60% energy from fat) × 8 wks	Microarray	[35]
	miR-199a-3p, miR-184 miR-203 miR-210 miR-383	repressed				
Oleic acid	miR-21	induced	HepG2 cells, human primary hepatocytes, Wistar rat liver tissues	Rats HFD: 19.5% fat (36.6% SFAs, 41.9% UFAs, 33.9% PUFAs), 50 μM on HepG2 cells and HPHs.	RT-qPCR	[36]
Stearic acid	miR-467b	repressed	Liver tissues, and hepatocytes, five wk old male C57BL/6J mice in two groups (*n* = 10), murine Hepa 1–6 hepatocytes	Mice fed HFD (20%lard) × 8 wks. In cells, saturated fatty acid-induced steatosis with 50 microM stearic acid SA × 24 h.	RT-qPCR.	[37]
Cholesterol, monounsaturated fatty acids (MUFAs), polyunsaturated fatty acids (PUFAs)	miR-122	repressed	Liver tissues, hepatocytes, Mini Pig Sus scrofa Standard diet (*n* = 5), High cholesterol diet (*n* = 7)	High cholesterol diet × 11 wks (22.77% crude fat, 19.3 MJ/kg, 2% cholesterol 0.55% MUFAs, 6.86% PUFAs)	qPCR	[38]
DHA	miR-141-3p, miR-221-3p, miR-30c, miR-192, miR-1283	induced	Caco-2, HepG2 hepatocytes	Fatty acids delivered to cells as lipid micelles with phosphatidylcholine × 24 h with 200 μM/L oleic and palmitic acid × 24 h.	Microarray, qRT-PCR	[39]
	miR-30a	repressed				
DHA + Palmitic fatty acid	miR-1	induced				
Palmitic fatty acid	miR-106b	induced				
DHA + Oleic Acid	let-7f, miR-181a-5p	induced				
Cholesterol	miR-151b	induced	Caco-2 cells, murine brain, cerebellum and kidney tissues. Mice and cell lines	In cells: cholesterol, CLA, and DHA delivered to cells as micelles with lysophosphatidylcholine and sodium taurocholate. In animals: 8wk-old C57BL6/J mice fed a normal chow diet or an HFD containing 1.25% of cholesterol × 16 wks after mice sacrificed and tissue samples collected.	RT-qPCR	[40]
	miR-215	repressed				
CLA (conjugated linoleic acid)	miR-224, miR-106b miR-16 miR-122 miR-151-3p miR-107 miR-151b	induced				
	miR-192, miR-215 miR-3141 miR-4739 miR-4534	repressed				
DHA	miR-23a, miR-1260b let-7i miR-30d miR-183 miR-92b miR-107 miR-320e miR-151b	induced				
	miR-192, miR-215 miR-4454 miR-4787-5p miR-3960 miR-4739 miR-3665 miR-3141 miR-3940-5p miR-4687-3p	repressed				
CLA 1	miR-143	induced	Mice, retroperitoneal adipose tissue	CLA 1 = Standard diet (SD)+ conjugated linoleic acids (CLA) 3 mg × 37 days, CLA 2 = SD + CLA 10 mg, CLA 3= HFD + CLA 6 mg, CLA 4= HFD + CLA 20 mg	qPCR	[41]
CLA 2, 4		repressed				
CLA 3		no change				
CLA 1, 2, 3	miR-103	induced				
CLA 4		repressed				
CLA 1, 2, 4	miR-107	repressed				
CLA 3		induced				
CLA 1, 2, 4	miR-221	induced				
CLA 3		no change				
CLA 1, 2, 3, 4	miR-222	induced				
CLA 2, 4 CLA 3	miR-328, miR-330-3p, miR-221, miR-125a-5p	repressed	Blood plasma. In the first study, 20 miRNAs were differentially expressed *n* = 10 healthy women, BMI 30–35. Validation study *n* = 20 (8 m, 12 w)	30 g almonds and walnuts × 8 wks (2.02 g n-3, 11.1 n-6) in a normocaloric diet.	Microarray, RT-qPCR	[42]
	miR-192, miR-486-5p, miR-19b, miR-106a, miR-18a, miR-130b	no change				
CLA 1, 2, 3	miR-103	induced	Serum, PBMCs, adipocytes, hepatocytes	AIN-93G rat chow mixed with lard and corn oil, plus EPA and DHA or omega-6 1(0 μL/100 g/day) × 16 wks	Microarray, RT-qPCR	[43]
CLA 4 CLA 1, 2, 4	miR-1286, miR-619-3p, miR-302c-5p, miR-519b-3p, miR-614, miR-23b-3p	repressed	Healthy women, PBMCs	50 mL (44 g) single dose 8 a.m., samples were taken after 4.	Microarray, RT-qPCR	[44]
	miR-107	repressed				
CLA 3	miR-96	induced	Mice. The liver and gastrocnemius skeletal muscle, Hep2 cells	HFD, 60% calories from fat × 14 weeks, cells treated with palmitate (0.5 mM) or oleate (0 ± 0.5 mM) for 18 h	RT-qPCR	[45]
CLA 1, 2, 4	miR-221	induced	HepG2 cell line	Palmitic, oleic acid (1:2) long-chain mixture different concentrations for 24 h.	RT-qPCR	[46]
CLA 3	miR-223, miR-135a	no change	Men, purified HDL from plasma	high in iTFA (10.2 g/2500 kcal, 3.7% daily energy), high in rTFA (10.2 g/2500 kcal, 3.7% daily energy), control diet low in TFAs (2.2 g/2500 kcal, 0.8% daily energy (each for four weeks, >3 week wash-out period)	RT-qPCR	[13]
CLA 1, 2, 3, 4	miR-222	induced	Men, purified HDL from plasma	High in iTFA (10.2 g/2500 kcal, 3.7% daily energy), high in rTFA (10.2 g/2500 kcal, 3.7% daily energy), control diet low in TFAs (2.2 g/2500 kcal, 0.8% daily energy (each for four weeks, >3 week wash-out period)	microarrays, RT-qPCR	[16]
iTFA (vaccenic acid 28 g/100 g) vs. Control	miR-199a-5p, miR-30a-5p	induced				
rTFA, iTFA vs. control	miR-328-3p, miR-423-3p, miR-124-3p, miR-150-5p, miR-31-5p, miR-375	repressed				
iTFA vs. rTFA	miR-133a-3p	repressed				

Abbreviations: C57BL6J, C57 black 6 mouse strain; Caco-2, human epithelial colorectal adenocarcinoma cells; CLA, conjugated acid linoleic; DHA, docosahexaenoic acid; EPA, eicosapentaenoic acid; HepG2, human hepatoma G2 cells; HDL, high-density lipoprotein cholesterol; HFD, high-fat diet; HPHs, human primary hepatocytes; iTFA, industrial trans fatty acids; PBMCs, peripheral blood mononuclear cells; rTFA, ruminant trans fatty acids; RT-qPCR, real time polymease chain reaction; SFAs, saturated fatty acids; wks, weaks.

## Data Availability

Not applicable.

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
