# Peer review of "Effect of Dietary Fatty Acids on MicroRNA Expression Related to Metabolic Disorders and Inflammation in Human and Animal Trials"

_nutrients, 2021, doi:10.3390/nu13061830_

Round 1

Reviewer 1 Report

The present review was designed for dietary fatty acids on microRNA expression related to metabolic disorders and inflammation in human and animal trials. The manuscript is well written and the results section is very much in details which I believe strengthen the manuscript. I appreciated the design of this study and I did not see any evident gaps. 

Author Response

Response to Reviewer 1 Comments

Point 1: The present review was designed for dietary fatty acids on microRNA expression related to metabolic disorders and inflammation in human and animal trials. The manuscript is well written and the results section is very much in details which I believe strengthen the manuscript. I appreciated the design of this study and I did not see any evident gaps. 

Response 1: Thank you for your comments.

Reviewer 2 Report

This is a narrative review of studies evaluating the effects of varying dietary fat sources on micro-RNA expression in cells and serum of humans, mammals, and cultured cells.  This is a challenging task, for 2 reasons.   First, there are at least 40 different dietary fatty acids commonly found in multiple classes based upon chain length, degree of unsaturation, positions of unsaturated bonds, and whether double bonds are in the cis- or trans-isomer configuration.  Second, there are myriad mi-RNA structures that may vary in their epigenetic effects based upon the specific tissue and ligand. 

In an understandable attempt at simplification, the authors have chosen to classify diets as rich in saturated, mono-unsaturated, and polyunsaturated fats; with sub classes for omega-6, omega-3, plus 3 versions of trans fats:  industrial, ruminant, and CLA.  Unfortunately, this reductionist approach severely compromises the validity and utility of this review.  Examples: 

  1. Saturated fatty acids are not uniform in their metabolic effects. Palmitate is understood to be far more atherogenic than stearate.
  2. There are no ‘pure’ natural dietary fats. They are all mixtures.  Lard is typically equal in its proportions of saturates and mono-unsaturates. 
  3. In addition to its high oleic acid content, extra virgin olive oil is rich in bioactive polyphenols, not to mention squalene. Anything attributed to its dietary intake as a mono-unsaturate effect is highly questionable. 
  4. Conjugated linoleic acid is part of the trans-fatty acid family, and it is derived from dairy trans fats. Thus these categories are indistinguishable. 
  5. Among the omega-6 PUFA, arachidonic acid is both the most prevalent in somatic membranes and the most bioactive of all omega-6 fatty acids, including its epigenetic effects on gene expression. It is also an important component of dietary fats of animal origin, found in red meats, poultry, fish, and eggs. 

Another concern with this review is the implicit presumption that fats consumed in the diet are necessarily quantitatively reflected in the tissue(s) of interest.  This is relatively true for cells grown in tissue culture, variably true for rapidly-growing juvenile rodents, and rarely true for adult humans fed a defined diet for less than 2 years. 

And finally, the authors promote the increasingly questionable conclusion that dietary saturates increase all biomarkers of CVD risk.  Copious evidence published over the last decade contravenes this hypothesis.  See:

  1. Siri-Tarino PW, et al. Am J Clin Nutr.  2010; 91:535-546.
  2. Astrup A, et al. J Am Col Cardiology.  2020;76:844-857

Furthermore, multiple human studies now demonstrate that high fat ketogenic diets – many enriched with saturated fats -- do not increase serum saturated fat content, and are not associated with increased inflammation.  These observations contravene many of the conclusions in this review.  E.g., Forsythe C, et al.  Lipids. 2008; 43:65-77.

Specific comments:

Lines 38-40.  Excessive consumption of DFA causes obesity, NAFLD, and cardiovascular disease.  The increasing accepted view is that atherogenic dyslipidemia in humans is more closely associated with the intake of simple sugars and refined carbohydrates.    

Lines 42-43.  As noted above, chronic metabolic diseases in humans are no longer considered  attributable to dietary saturated fat intake. 

Lines 55-58.  This is only true for the calculated LDL-cholesterol.  HDL-C, small dense LDL, and serum triglycerides do not show a beneficial response with increased dietary PUFA. 

Lines 62-72.  Trans fatty acids contain trans double bonds.  These are not created by hydrogenation per se.  They are the secondary products of cis-double bonds interacting with the catalysts used to partially hydrogenate high PUFA vegetable oils. 

Line 79.  miRNAs are not ‘genes’.  They are the RNA products of transcription of non-coding segments of the genome. 

Lines 183-185.  Lard is typically approx. 50% mono-unsaturated. 

Lines 235-245.  Authors cite Ref #35 that DHA induces miR-107.  They also claim that miR-107 binds to the GLOCK gene, disrupting circadian rhythm, and this is associated with T2D, ASCVD, and obesity.  Is this just another random association?  If not, where is the evidence that DHA or fish oil cause or exacerbate these disease states?

Lines 258-259.  EVOO is a complex nutrient mix that is rich in polyphenols and squalene.  Any conclusions as to the fatty acid effects of EVOO on miRNA are fraught at best. 

Lines 282-302.  Suffers from a lack of differentiation between industrial TFAs and ruminant TFAs.

Lines 314-321.  HepG2 cells treated with the same fatty acid mix seems to have simultaneously inhibited and over-expressed miR-26a in these cells.  Perhaps there is important information to be gleaned from this discussion, but this paragraph defies understanding. 

Lines 325-330.  These conclusions seem to reflect the dietary views of the author’s, but the many prior pages of results and discussion are confusing at best, if not obfuscatory. 

Author Response

Response to Reviewer 2 Comments

Point 1: This is a narrative review of studies evaluating the effects of varying dietary fat sources on micro-RNA expression in cells and serum of humans, mammals, and cultured cells.  This is a challenging task, for 2 reasons.   First, there are at least 40 different dietary fatty acids commonly found in multiple classes based upon chain length, degree of unsaturation, positions of unsaturated bonds, and whether double bonds are in the cis- or trans-isomer configuration.  Second, there are myriad mi-RNA structures that may vary in their epigenetic effects based upon the specific tissue and ligand. 

In an understandable attempt at simplification, the authors have chosen to classify diets as rich in saturated, mono-unsaturated, and polyunsaturated fats; with sub classes for omega-6, omega-3, plus 3 versions of trans fats:  industrial, ruminant, and CLA.  Unfortunately, this reductionist approach severely compromises the validity and utility of this review.  Examples: 

  1. Saturated fatty acids are not uniform in their metabolic effects. Palmitate is understood to be far more atherogenic than stearate.
  2. There are no ‘pure’ natural dietary fats. They are all mixtures.  Lard is typically equal in its proportions of saturates and mono-unsaturates. 
  3. In addition to its high oleic acid content, extra virgin olive oil is rich in bioactive polyphenols, not to mention squalene. Anything attributed to its dietary intake as a mono-unsaturate effect is highly questionable. 
  4. Conjugated linoleic acid is part of the trans-fatty acid family, and it is derived from dairy trans fats. Thus these categories are indistinguishable. 
  5. Among the omega-6 PUFA, arachidonic acid is both the most prevalent in somatic membranes and the most bioactive of all omega-6 fatty acids, including its epigenetic effects on gene expression. It is also an important component of dietary fats of animal origin, found in red meats, poultry, fish, and eggs. 

Response 1: As you comment, we agree that is difficult to distinguish diets rich in a specific kind of faty acid. However, the purpose of our study was to clarify the possible effects of different types of faty acids in the regulation at molecular level, such as in the expression of miRNAs in metabolism. In fact, the studies performed have the weakness of this simplification.  A good example as you mention is the study by Merel and cols. (2020), related to palmitic acid vs stearic acids as markers in the cardiometabolic risk, also clearly presented this weakness. 

We included in the new version of the manuscript all comments about limitations of previous studies, attempting to explain the effects of specific diets in human metabolism and at epigenetic level.  We think that now, it is more evident that this type of limitations is found in the studies performed previously. However, our review indicates that diets are able to modify genes at epigenetic level through the expression of miRNAs involved in metabolism, by both the intervention of simplified diets, as well as in complex diets including various types of fatty acids. The references cited by you were included and discussed.

Merel A. et al., Palmitic Acid Versus Stearic Acid: Effects of Interesterification and Intakes on Cardiometabolic Risk Markers—A Systematic Review Nutrients. 2020 Mar; 12(3): 615.

Point 2: Another concern with this review is the implicit presumption that fats consumed in the diet are necessarily quantitatively reflected in the tissue(s) of interest.  This is relatively true for cells grown in tissue culture, variably true for rapidly-growing juvenile rodents, and rarely true for adult humans fed a defined diet for less than 2 years. 

Response 2:

Indeed, tissue responses depends of the nutrients milieu. This point has been clarified in the graphical abstract as well as in the manuscript.

Point 3: And finally, the authors promote the increasingly questionable conclusion that dietary saturates increase all biomarkers of CVD risk.  Copious evidence published over the last decade contravenes this hypothesis.  See:

  1. Siri-Tarino PW, et al. Am J Clin Nutr.  2010; 91:535-546.
  2. Astrup A, et al. J Am Col Cardiology.  2020;76:844-857

Furthermore, multiple human studies now demonstrate that high fat ketogenic diets – many enriched with saturated fats -- do not increase serum saturated fat content, and are not associated with increased inflammation.  These observations contravene many of the conclusions in this review.  E.g., Forsythe C, et al.  Lipids. 2008; 43:65-77.

Response 3: According to your suggestions, we included in the new version of the manuscript all comments about limitations of previous studies, attempting to explain the effects of specific diets in human metabolism and at epigenetic level.  We think that now, it is more evident that these types of limitations are found in the studies performed previously. However, our review indicates that diets are able to modify genes at a epigenetic level through the expression of miRNAs involved in metabolism, by both the intervention of simplified diets, as well as in complex diets including various types of fats.

The references cited by the reviewer were included and discussed.

Specific comments:

Lines 38-40.  Excessive consumption of DFA causes obesity, NAFLD, and cardiovascular disease.  The increasing accepted view is that atherogenic dyslipidemia in humans is more closely associated with the intake of simple sugars and refined carbohydrates.    

Response: The phrase was changed and the following sentence was included: On the other hand, it is also well known that atherogenic dyslipidaemia is closely associated with the intake of simple sugars and refined carbohydrates, which had been previously studied in depth [4,5]⁠ (Lines 43-45).  

Lines 42-43.  As noted above, chronic metabolic diseases in humans are no longer considered  attributable to dietary saturated fat intake. 

Response: This phrase was omitted.

Lines 55-58.  This is only true for the calculated LDL-cholesterol.  HDL-C, small dense LDL, and serum triglycerides do not show a beneficial response with increased dietary PUFA. 

Response: The phrase was changed and this sentence was included:

However, the substitution of SFAs with polyunsaturated fatty acids (PUFAs) in the diet appears to have favorable effects on LDL-cholesterol (lines 55-57).

Lines 62-72.  Trans fatty acids contain trans double bonds.  These are not created by hydrogenation per se.  They are the secondary products of cis-double bonds interacting with the catalysts used to partially hydrogenate high PUFA vegetable oils. 

Response: We have included a new paragraph to clarify this point:

“Moreover, TFAs are another type of fatty acid formed industrially, that also contain double bonds. They are secondary products of cis-double bonds interacting with certain catalysts used to partially hydrogenate high PUFA vegetable oils. Ruminant TFAs are also found in ruminant meat and milk resulting from the biohydrogenation of unsaturated fatty acids by bacterial metabolism found in the rumen of animals like cattle and goats [6,9,11]” (lines 68-74)⁠.

Line 79.  miRNAs are not ‘genes’.  They are the RNA products of transcription of non-coding segments of the genome. 

Response: We agree with you that the mature miRNAs are not genes and to avoid confusions we included a new explanation about that. There are miRNA genes and there are products of the transcription of these genes that are named miRNAs. For example, mir-82 is the name of the gene for miRNA-82 and miRNA-82 is the product.  Indeed, actually the concept for miRNAs genes described at less two types: one that possess a promotor region specific for its transcription and others that are generated by splicing of coded genes.  However, to avoid confusion the phrase was changed (lines 82-83).

Lines 183-185.  Lard is typically approx. 50% mono-unsaturated. 

Response:  The phrase, “In a high cholesterol diet (HCD) with lard for 11 weeks”  was changed to “in a diet with lard for 11 weeks” (line 192).

Lines 235-245.  Authors cite Ref #35 that DHA induces miR-107.  They also claim that miR-107 binds to the GLOCK gene, disrupting circadian rhythm, and this is associated with T2D, ASCVD, and obesity.  Is this just another random association?  If not, where is the evidence that DHA or fish oil cause or exacerbate these disease states?

Response:  As you comment, indeed, treatment with DHA induces the expression of miRNA-107, which involved in the Clock gene regulation. Thus, its inhibition is detrimental for circadian rhythm representing a possible therapeutic target in metabolic disease such as Type-2 Diabetes, and ASCVD. The sentence was rewritten in the new manuscript (lines 242-246).

Lines 258-259.  EVOO is a complex nutrient mix that is rich in polyphenols and squalene.  Any conclusions as to the fatty acid effects of EVOO on miRNA are fraught at best. 

Response: EVOO has the following major components, please see Table 2: Major EVOO components from Jimenez-Lopez C., et al. Bioactive Compounds and Quality of Extra Virgin Olive Oil, Foods. 2020 Aug; 9(8): 1014.

As MUFAs conform 65.2–80.8% of EVOO compostion, we feel there is reason to believe EVOO has effects on miRNA expression. We included this sentence in the new manuscript:

“EVOO is a complex nutrient mix composed primarily of MUFAs (65.2–80.8%), MUFAs (7.0-15.5%), tocopherols, the carbohyadrate squalene, pigments and other compounds such as phenols, triterpene dialcohols and B-sitosterol [45]⁠” (lines 261-263).

Lines 282-302.  Suffers from a lack of differentiation between industrial TFAs and ruminant TFAs.

Response: We complement the sentences as follows:

“rTFA, (naturally occurring in ruminant meat, and milk fat)” (lines 287-288)

“(iTFAs, produced industrially by partial hydrogenation of vegetable oils)” (lines 291-292)

Lines 314-321.  HepG2 cells treated with the same fatty acid mix seems to have simultaneously inhibited and over-expressed miR-26a in these cells.  Perhaps there is important information to be gleaned from this discussion, but this paragraph defies understanding. 

Response: We rewrote this paragraph carefully to make it understandable, thank you very much (lines 321-328).

Lines 325-330.  These conclusions seem to reflect the dietary views of the author’s, but the many prior pages of results and discussion are confusing at best, if not obfuscatory. 

Response: According to your comment, this paragraph was rewritten to avoid confusions (lines 321-328).

Reviewer 3 Report

This review provides a good picture of published data about microRNA expression and dietary fatty acids. Only minor correction could be useful.

Table one, and text provide relevant descriptive observations of available experimental results, but paper would be greatly improved discussing information about possible integrate conclusions. For example, it would be very interesting to have a discussion about miRNA species  resulting similarly regulated in multiple experimental systems, if there are any. This would allow to better evaluate the general relevance of the data reported in the literature.

LIne 36 and line 37 seem to indicate that polyunsaturated fatty acids are usually one-third of  total energy intake and this could be a mistyping. Minor English correction are necessary.

Author Response

Response to Reviewer 3 Comments

Point 1: This review provides a good picture of published data about microRNA expression and dietary fatty acids. Only minor correction could be useful.

Table one, and text provide relevant descriptive observations of available experimental results, but paper would be greatly improved discussing information about possible integrate conclusions. For example, it would be very interesting to have a discussion about miRNA species  resulting similarly regulated in multiple experimental systems, if there are any. This would allow to better evaluate the general relevance of the data reported in the literature.

Line 36 and line 37 seem to indicate that polyunsaturated fatty acids are usually one-third of  total energy intake and this could be a mistyping. Minor English correction are necessary.

Response 1:

The mistyping error about total energy intake has been corrected (lines 39-40) and the English has been revised.

Reviewer 4 Report

In this interesting review, the authors discuss the effect of dietary fatty acids on microRNA expression related to metabolic disorders and inflammation in human and animal trials from last 10 years. First, they address the detailed description of different dietary fatty acids and their role in obesity and cardiovascular diseases. In further parts, they introduce microRNAs (miRNAs) and discuss the mechanisms regulating obesity and cardiovascular diseases. Finally, they discuss the key findings from last 10 years about the effects of DFAs on miRNA expression.

This is a very well written review, which gives us a bright overview over the relationship between DFAs and miRNA and obesity and cardiovascular diseases.

I have only a minor remark, which I believe is important for the completion of this review. The authors introduced the importance of LDL in regulation of miRNAs in obesity and cardiovascular diseases. They should extend this part with oxLDL since there are many publications (PMID: 33593717, PMID: 33593717, PMID: 33470395) showing the importance of oxLDL induced miRNA dependent regulation in obesity and cardiovascular diseases.

Author Response

Response to Reviewer 4 Comments

Point 1:

In this interesting review, the authors discuss the effect of dietary fatty acids on microRNA expression related to metabolic disorders and inflammation in human and animal trials from last 10 years. First, they address the detailed description of different dietary fatty acids and their role in obesity and cardiovascular diseases. In further parts, they introduce microRNAs (miRNAs) and discuss the mechanisms regulating obesity and cardiovascular diseases. Finally, they discuss the key findings from last 10 years about the effects of DFAs on miRNA expression.

This is a very well written review, which gives us a bright overview over the relationship between DFAs and miRNA and obesity and cardiovascular diseases.

I have only a minor remark, which I believe is important for the completion of this review. The authors introduced the importance of LDL in regulation of miRNAs in obesity and cardiovascular diseases. They should extend this part with oxLDL since there are many publications (PMID: 33593717, PMID: 33593717, PMID: 33470395) showing the importance of oxLDL induced miRNA dependent regulation in obesity and cardiovascular diseases.

Response 1:

Thank you for your comments and suggestions.

The following phrase, and references PMID: 33593717 and PMID: 33470395, were added.

“In addition, it has been reported that DFAs and lipoproteins such as ox-LDL (oxidized low-density lipoprotein) induce miRNA depend regulation in obesity and CVD [15,16]⁠” (lines 81-83)

Round 2

Reviewer 2 Report

From the prior review:

This reviewer’s specific comment on the lack of risks associated with dietary saturated fats on cardiovascular risk and inflammation. 

Furthermore, multiple human studies now demonstrate that high fat ketogenic diets – many enriched with saturated fats -- do not increase serum saturated fat content, and are not associated with increased inflammation. These observations contravene many of the conclusions in this review. E.g., Forsythe C, et al. Lipids. 2008; 43:65-77.

Response 3: According to your suggestions, we included in the new version of the manuscript all comments about limitations of previous studies, attempting to explain the effects of specific diets in human metabolism and at epigenetic level. We think that now, it is more evident that these types of limitations are found in the studies performed previously. However, our review indicates that diets are able to modify genes at a epigenetic level through the expression of miRNAs involved in metabolism, by both the intervention of simplified diets, as well as in complex diets including various types of fats.  The references cited by the reviewer were included and discussed.

Unfortunately, the discussion of dietary fatty acids and inflammation was not adequately revised and the suggested reference (one of many) illustrating that high fat and ketogenic diets in humans actually reduce inflammation was not included. 

Specific comments: 

Lines 43-45.  New references 4 and 5 address that lack of risk associated with dietary saturated fatty acids.  However these references do not specifically address the atherogenic dyslipidemia and cardiovascular risks associated with dietary simple sugars and refined carbohydrates.   

Lines 48, and lines 51-53.  TFA (trans-fatty acids) needs to be defined in this manuscript before the acronym is used.

Lines 50-51.  Serum LDL cholesterol is not an accepted risk factor for developing obesity.  Association does not equal causality. 

Lines 55-57.  In humans, increased dietary PUFA also decreases serum HDL-cholesterol in many published studies lasting longer than a few weeks.  Sadly, another example of the authors’ systemic bias against dietary saturated fats as called out in the prior review.

Table 1.  Despite prior review comments, this remains dietarily incorrect.  Examples:

Section PUFA-EPA.  “mice fed palm oil, fish oil, olive oil”.  Fish oil is typically about 30% long-chain omega-3, and only half of that is EPA.  Thus this was a mixed fatty acid diet in which a minority of the animal’s fatty intake was EPA.

Section SFA, SFA.  The mice were fed 60% of dietary energy as lard.  The majority of fatty acids in typical lard is composed of mono-unsaturates and polyunsaturates.  Thus a minority of the dietary fat in this study came from saturated fatty acids. 

SFA, Stearic .  Mice were fed 20% of energy as lard for 8 weeks.  Lard is less that 40% saturates, and much of that fraction is palmitate, not Stearic acid. 

Lines 160-162.  The results showed that an excess of circulating DFAs induced miRNA-21  expression which downregulates PTEN expression and contributes to the development of liver steatosis [33].

Comment:  Reference 33 states:  Phosphatase and tensin homolog (PTEN) is a regulator of phosphoinositide 3‐kinase signaling and an important tumor suppressor mutated/deleted in human cancers. PTEN deletion in the liver leads to insulin resistance, steatosis, inflammation, and cancer. We recently demonstrated that unsaturated fatty acids trigger steatosis by down‐regulating PTEN expression in hepatocytes via activation of a mammalian target of rapamycin (mTOR)/nuclear factor kappa B (NF‐κB) complex….  In contrast, unsaturated fatty acids up‐regulate the expression of microRNA‐21, which binds to PTEN messenger RNA 3′‐untranslated region and induces its degradation. 

This study clearly indicates that dietary unsaturates are harmful, which contravenes the authors’ conclusions. 

Lines 251-255.  In a subsequent study, ten miRNAs were validated in both men and women in a larger group. Changes were seen in miRNA expression after eight weeks with daily walnut and almond intake. Importantly, changes were associated with DFAs and inflammatory biomarkers.

Comment:  The authors seem to be attributing any changes in inflammation biomarkers to DFAs, which is questionable.  Walnuts and almonds are rich in phytonutrients and fiber, both of which are associated with reduced inflammation.  In particular, intestinal butyrate production from dietary fiber has potent epigenetic effects that mitigate oxidative stress and inflammation. 

Lines 262.  Squalene is the final metabolic intermediate in the hepatic synthesis of cholesterol from acetate.  Squalene is not a carbohydrate.  

Lines 364-67.  Altogether, these findings suggest that SFAs in the diet are capable of modulating miRNA expression related to increased inflammation and obesity, whereas PUFAs in the diet, particularly EPA and DHA, seem to do the opposite.  

Comment:  This is an overly reductionist view of clinical reality.  See for example Forsythe et al 2008 – mentioned in the prior review.

Lines 450-451.  Additionally, unsaturated FAs generally exert a positive effect on metabolism, while SFAs and TFAs exert a negative one. 

Comment:  See comments on ref #33 above.  Also note that conjugated linoleic acid is a trans fatty acid that the authors extoll as beneficial, but do not adequately differentiate it from other trans fatty acids despite this reviewer’s stated concerns in a prior review. 

Author Response

Response to Reviewer 2 Comments (Round 2)

Point 1: From the prior review:

This reviewer’s specific comment on the lack of risks associated with dietary saturated fats on cardiovascular risk and inflammation. 

Furthermore, multiple human studies now demonstrate that high fat ketogenic diets – many enriched with saturated fats -- do not increase serum saturated fat content, and are not associated with increased inflammation. These observations contravene many of the conclusions in this review. E.g., Forsythe C, et al. Lipids. 2008; 43:65-77.

Unfortunately, the discussion of dietary fatty acids and inflammation was not adequately revised and the suggested reference (one of many) illustrating that high fat and ketogenic diets in humans actually reduce inflammation was not included. 

Response 1:

According to your comment new references and discussion about the complexity of physiological responses in the organisms to dietary fatty acids in a normal diet, both by unsaturated and saturated fat intake. It is well known that in diets both types of lipids are present, for example, milk contains frequently more saturated fatty acids than unsaturated fatty acids. These comments are in the abstract, text, and conlcusions.

Specific comments: 

Lines 43-45.  New references 4 and 5 address that lack of risk associated with dietary saturated fatty acids.  However these references do not specifically address the atherogenic dyslipidemia and cardiovascular risks associated with dietary simple sugars and refined carbohydrates.   

 Response:

Four new references, in addition to the three suggested by the reviewer, were all included concerning this subject.  We included comments about CVD and metabolic risk from the consumption of diets containing dietary simple sugars and refined carbohydrates.

Lines 48, and lines 51-53.  TFA (trans-fatty acids) needs to be defined in this manuscript before the acronym is used.

 Response: The first time the acronym (TFAs) was used it was defined.

Lines 50-51.  Serum LDL cholesterol is not an accepted risk factor for developing obesity.  Association does not equal causality. 

Response: As you comment, the association of serum LDL cholesterol with obesity was mentioned, but not as for its causality. 

Lines 55-57.  In humans, increased dietary PUFA also decreases serum HDL-cholesterol in many published studies lasting longer than a few weeks.  Sadly, another example of the authors’ systemic bias against dietary saturated fats as called out in the prior review.

Response: In the new version of the manuscript care has been taken to avoid a bias against dietary saturated fats, as we mention in the previous paragraphs.

Table 1.  Despite prior review comments, this remains dietarily incorrect.  Examples:

Section PUFA-EPA.  “mice fed palm oil, fish oil, olive oil”.  Fish oil is typically about 30% long-chain omega-3, and only half of that is EPA.  Thus this was a mixed fatty acid diet in which a minority of the animal’s fatty intake was EPA.

Section SFA, SFA.  The mice were fed 60% of dietary energy as lard.  The majority of fatty acids in typical lard is composed of mono-unsaturates and polyunsaturates.  Thus a minority of the dietary fat in this study came from saturated fatty acids. 

SFA, Stearic .  Mice were fed 20% of energy as lard for 8 weeks.  Lard is less that 40% saturates, and much of that fraction is palmitate, not Stearic acid. 

Response: The column (section) was eliminated to avoid confusion.  

Lines 160-162.  The results showed that an excess of circulating DFAs induced miRNA-21  expression which downregulates PTEN expression and contributes to the development of liver steatosis [33].

Comment:  Reference 33 states:  Phosphatase and tensin homolog (PTEN) is a regulator of phosphoinositide 3‐kinase signaling and an important tumor suppressor mutated/deleted in human cancers. PTEN deletion in the liver leads to insulin resistance, steatosis, inflammation, and cancer. We recently demonstrated that unsaturated fatty acids trigger steatosis by down‐regulating PTEN expression in hepatocytes via activation of a mammalian target of rapamycin (mTOR)/nuclear factor kappa B (NF‐κB) complex….  In contrast, unsaturated fatty acids up‐regulate the expression of microRNA‐21, which binds to PTEN messenger RNA 3′‐untranslated region and induces its degradation. 

This study clearly indicates that dietary unsaturates are harmful, which contravenes the authors’ conclusions. 

Response: This paragraph was rewritten and modified according to your comments. 

Lines 251-255.  In a subsequent study, ten miRNAs were validated in both men and women in a larger group. Changes were seen in miRNA expression after eight weeks with daily walnut and almond intake. Importantly, changes were associated with DFAs and inflammatory biomarkers.

Comment:  The authors seem to be attributing any changes in inflammation biomarkers to DFAs, which is questionable.  Walnuts and almonds are rich in phytonutrients and fiber, both of which are associated with reduced inflammation.  In particular, intestinal butyrate production from dietary fiber has potent epigenetic effects that mitigate oxidative stress and inflammation. 

Response: The paragraph was changed to,

The effects of PUFAs in miRNA expression were assessed in a healthy population. The effect of DFAs in a PUFA-enriched diet, inferred from plasma fatty acid concentration, was linked to changes in circulating miRNAs. In the first experiment, 20 miRNAs were identified and differentially expressed in healthy women after consuming PUFAs in their diet. In a subsequent study, ten miRNAs were validated in both men and women in a larger group. In fact, changes were seen in miRNA expression after eight weeks with daily walnut and almond intake [44]⁠. In particular, miR-221 and CRP (High-sensitivity C-reactive protein) expression were repressed after treatment with PUFAs.

Lines 262.  Squalene is the final metabolic intermediate in the hepatic synthesis of cholesterol from acetate.  Squalene is not a carbohydrate.  

 Response: This was a mistake and was corrected.  

Lines 364-67.  Altogether, these findings suggest that SFAs in the diet are capable of modulating miRNA expression related to increased inflammation and obesity, whereas PUFAs in the diet, particularly EPA and DHA, seem to do the opposite.  

Comment:  This is an overly reductionist view of clinical reality.  See for example Forsythe et al 2008 – mentioned in the prior review.

Response: The Forsythe reference was included and discussed according to your comment. We make efforts to be not reductionist as you mention. The paragraph was rewritten. 

Lines 450-451.  Additionally, unsaturated FAs generally exert a positive effect on metabolism, while SFAs and TFAs exert a negative one. 

Comment:  See comments on ref #33 above.  Also note that conjugated linoleic acid is a trans fatty acid that the authors extoll as beneficial, but do not adequately differentiate it from other trans fatty acids despite this reviewer’s stated concerns in a prior review. 

Response: The conclusion has been revised and corrected according to your comment.
